# FINDING SHARED DECODABLE CONCEPTS AND THEIR NEGATIONS IN THE BRAIN

**Cory Efird**
Computing Science
University of Alberta
Alberta Machine Intelligence Institute (Amii)
efird@ualberta.ca

**Alex Murphy**
Computing Science
University of Alberta
Alberta Machine Intelligence Institute (Amii)
amurphy3@ualberta.ca

**Joel Zylberberg**
Physics and Astronomy
York University
joelzy@yorku.ca

**Alona Fyshe**
Computing Science and Psychology
University of Alberta
Alberta Machine Intelligence Institute (Amii)
alona@ualberta.ca

## ABSTRACT

Prior work has offered evidence for functional localization in the brain; different anatomical regions preferentially activate for certain types of visual input. For example, the fusiform face area preferentially activates for visual stimuli that include a face. However, the spectrum of visual semantics is extensive, and only a few semantically-tuned patches of cortex have so far been identified in the human brain. Using a multimodal (natural language and image) neural network architecture (CLIP, Radford et al. (2021)), we train a highly accurate contrastive model that maps brain responses during naturalistic image viewing to CLIP embeddings. We then use a novel adaptation of the DBSCAN clustering algorithm to cluster the parameters of these participant-specific contrastive models. This reveals what we call Shared Decodable Concepts (SDCs): clusters in CLIP space that are decodable from common sets of voxels across multiple participants.

Examining the images most and least associated with each SDC cluster gives us additional insight into the semantic properties of each SDC. We note SDCs for previously reported visual features (e.g. orientation tuning in early visual cortex) as well as more complex concepts such as faces, places and bodies. We also uncover previously unreported areas with visuo-semantic sensitivity such as regions of extrastriate body area (EBA) tuned for legs/hands and sensitivity to numerosity in right intraparietal sulcus, sensitivity associated with visual perspective (close/far) and more. Thus, our contrastive-learning methodology better characterizes new and existing visuo-semantic representations in the brain by leveraging multimodal neural network representations and a novel adaptation of clustering algorithms.

## 1 INTRODUCTION

The study of the visual system has revealed a significant amount of knowledge about how the brain processes and organizes visual information. The early visual system has been well characterized for its ability to extract low level visual features (e.g. lines of different orientations). However, the picture becomes more murky as one moves beyond the early visual system. There is evidence for functional localization wherein large patches of cortex activate for specific visuo-semantic properties: areas of the brain preferentially activate for visual stimuli containing faces (Kanwisher et al., 1997), places (Epstein and Kanwisher, 1998), bodies (Orlov et al., 2010; Weiner and Grill-Spector, 2011; Bracci et al., 2015; 2010). There have been multiple recent attempts to identify new functionally localized regions of cortex with some converging evidence; for example, a food area near PPA has been identified (Khosla et al., 2022). So, while there has been progress towards parcelating the functional regions of the brain, only a few such broad visuo-semantic categories have been identified

and they typically cover fairly large areas of cortex. This leaves open the possibility that we may be able to detect new visuo-semantic categories in the brain or refine existing categories, and that these categories or sub-categories may become more apparent with a data-driven approach that can combine data across multiple participants.

But combining brain responses across participants is not simple. Prior research typically focused on mapping to a shared latent space (e.g. during stimulus reconstruction) for which spatial localization in the brain is not required. In cases where spatial information is required, analyses are typically applied to data which has been warped to a common template brain, often losing the connection to the unique structure of each participant's cortical layout. Prior work using clustering specifically of fMRI data is well represented in studies such as Golland et al. (2008); Lashkari et al. (2010); Thomas Yeo et al. (2011), though alternative methods based on independent component analysis (ICA) have also helped to advance understanding of fMRI selectivity (Norman-Haignere et al., 2015).

We present a novel method that finds semantically meaningful shared voxel clusters while still considering the voxels in each participant's native brain space, allowing us to sidestep the voxel alignment problem. Derived voxel clusters can be mapped to a shared space for visualization after identification of shared semantic clusters.

Our data-driven analysis of fMRI recordings during image-viewing allows not only for identification of the images that *most* drive neural activity in a given area, but also of images which are most associated with below-average BOLD response. This provides stronger evidence for the possible function of cortical areas, in some cases refining the hypotheses one might draw from a set of images. Brain areas showing below average in response to a stimuli may be consistent with recently reported "offsembles" of neurons (Pérez-Ortega et al., 2024) which show inhibited activity in response to e.g. visual stimuli, implying that, when interpreting neural representations, below average activity may be as important as above average.

We identify shared decodable concepts (SDCs) that drive activity in patches of cortex across multiple participants. We find that these SDCs are semantically interpretable and that their localization is fairly consistent across participants. We identify SDCs with our novel variant of DBSCAN that combines data from multiple fMRI participants, specifically searching for SDCs present in multiple subjects.

## 2 Decoding CLIP-Space from Brain Images

To identify shared decodable concepts (SDCs) in the brain, we derived a mapping from anatomical brain space to a representational space in which we can explore semantic sensitivity to visual input. In this section we describe the components of this mapping: a dataset of fMRI recordings (NSD), a multimodal image-text embedding model (CLIP), and our method to map from per-image brain responses to their associated multimodal embeddings. We consider two decoding models in this section, and verify that our proposed contrastive decoder outperforms a baseline ridge regression model.

### 2.1 fMRI Data

The natural scenes dataset (NSD) is a massive fMRI dataset acquired to study the underpinnings of natural human vision (Allen et al., 2022). Eight participants were presented with 30,000 images (10,000 unique images over 3 repetitions) from the Common Objects in Context (COCO) naturalistic image dataset (Lin et al., 2014). A set of 1,000 shared images were shown to all participants, while the other 9,000 images were unique to each participant. Single-trial beta weights were derived from the fMRI time series using the GLMSingle toolbox (Prince et al., 2022). This method fits numerous haemodynamic response functions (HRFs) to each voxel, as well as an optimized denoising technique and voxelwise fractional ridge regression, specifically for single-trial fMRI acquisition paradigms. Some participants did not complete all fMRI recording sessions and three sessions were held out by the NSD team for the Algonauts challenge. Further details can be found in Allen et al. (2022).

### 2.2 Representational Space for Visual Stimuli

To generate representations for each stimulus image, we use CLIP (Radford et al., 2021), a model trained on over 400 million text-image pairs with a contrastive language-image pretraining objective.

CLIP consists of a text-encoder and image-encoder that jointly learns a shared low-dimensional space trained to maximize the cosine similarity of corresponding text and image embeddings. We use the 32-bit Transformer model (ViT-B/32) implementation of CLIP to create a 512-dimensional representation for each stimulus image used in the NSD experiment. We train a decoder to map from fMRI responses during image viewing to the associated CLIP vector for that same image. Notably, because CLIP is a joint image-language model, these CLIP vectors also correspond to text captions in the pretraining stage, which can be used to describe the images presented to the participants.

## 2.3 DATA PREPARATION

**Data Split**   We split the per-image brain responses $X$ and CLIP embeddings $Y$ into training $(X_{\text{Train}}, Y_{\text{Train}})$, validation $(X_{\text{Val}}, Y_{\text{Val}})$, and test $(X_{\text{Test}}, Y_{\text{Test}})$ folds for each of the 8 NSD participants. For each participant, the validation and test folds were chosen to have exactly 1,000 images with three presentations. [1] Of the shared 1,000 images that all participants saw, 413 images were shown three times to every participant across the sessions released by NSD. These 413 images appear in each participant's testing fold.

**Voxel Selection**   The NSD fMRI data comes with voxelwise noise ceiling (NC) estimates that can be used to identify reliable voxels. However, the NC is calculated using the *full dataset*. Therefore, these NC estimates should not be used to extract a subset of voxels for decoding analyses because they are calculated using images in the test set, which is a form of double-dipping (Kriegeskorte et al., 2009). We therefore re-calculated the per-voxel noise ceiling estimates specifically on our designated training data only. We selected voxels with noise ceiling estimates above $8\%$ variance explainable to use as inputs to the decoding model. This resulted in 6k-19k voxel subsets per participant (see supplementary section A.2 for exact per-participant voxel dimensions input into the decoder) which were normalized according to section A.6. In our visualizations, regions of the flattened brain surface in black represent voxels that have passed this voxel selection threshold.

## 2.4 DECODING METHODOLOGY

**Decoding Model**   The decoding model $g(X; \theta) = \hat{Y}$ is a linear model trained to map brain responses $X = [x_1, \dots, x_n], x_i \in \mathbb{R}^v$ to brain embeddings $\hat{Y} = [y_1, \dots, y_n], y_i \in \mathbb{R}^{512}$. Here $n$ is the number of training instances, and $v$ is the participant-wise number of voxels that pass the noise ceiling threshold. We optimize the brain decoder using the InfoNCE definition of contrastive loss (van den Oord et al., 2018), which is defined below in Equations 1 and 2.

$$\text{Contrast}(A, B) = -\frac{1}{M} \sum_{i=1}^{M} log\left( \frac{exp(a_i \cdot b_i / \tau)}{\sum_{j=1}^{M} exp(a_i \cdot b_j / \tau)} \right) \tag{1}$$

$$\mathcal{L}_{\text{InfoNCE}}(A, B) = \frac{1}{2}[\text{Contrast}(A, B) + \text{Contrast}(B, A)] \tag{2}$$

In this definition $A = [a_1, \dots, a_n]$ and $B = [b_1, \dots, b_n]$ are embeddings for two modalities representing the same data points, the $\cdot$ operator represents cosine similarity, and $\tau$ is a temperature hyper-parameter. The loss is minimized when the distance between matching embeddings is small, and the distance between mismatched embeddings is large. In the original CLIP setting, $A$ and $B$ represent embeddings for images and corresponding text captions. In our implementation, we apply the contrastive loss to image embeddings computed by a pretrained frozen CLIP model and CLIP embeddings predicted from fMRI, i.e. we optimize $\min_\theta \mathcal{L}(\hat{Y}, Y_{\text{CLIP}}) = \mathcal{L}(g(X; \theta), Y)$. An illustration of the decoding procedure is given in Figure 11 (additional details on the decoding procedure are given in Section A.2).

**Evaluation**   We evaluate our models using top-k accuracy. Figure 12 shows the results of this evaluation. The contrastive decoder outperforms ridge regression across all values of $k$. Recall that

---

[1] Some participants in the NSD did not complete all scanning sessions and only viewed certain images once or twice. These images are assigned to the training set, which varies in size across participants.

our end goal is to identify shared decodable concepts (SDC) in the brain. Our methodology for this task relies on the *predicted* CLIP vectors, and so our subsequent analyses require an accurate decoding model.

# 3 FINDING SHARED BRAIN-DECODABLE CONCEPTS AND THEIR REPRESENTATIVE IMAGES

In this section, we describe our methods for finding and interpreting areas of the brain that respond to semantically similar images, and that are consistent across participants. We approach this problem by analyzing the weight matrices $W^{(k)}$ from the optimized linear decoders described in section 3.4 $g^{(k)}(x^{(k)}) = W^{(k)}x^{(k)} = y$ for all subjects $k \in \{1 \dots 8\}$. First, we observe that the decoder's linear transformation $W^{(k)}x^{(k)} = \sum_{i=1}^{v} w_i^{(k)} \cdot x_i^{(k)}$ is simply a summation of parameter vectors $w_i^{(k)} \in \mathbb{R}^{512}$ that are scaled by brain response values $x_i^{(k)} \in \mathbb{R}$. This means that when $x_i^{(k)}$ is above or below baseline activation, the CLIP dimension represented by $w_i^{(k)}$ is increased or decreased in the brain-decoded embedding. This motivates the key to our analysis: we view the parameter vector $w_i^{(k)}$ as a CLIP vector that represents a brain-decodable concept for voxel $i$. This allows us to use cosine distance $d(w_i^{(k)}, w_j^{(r)})$ between the weight vectors for voxels $i, j$ from participants $k, r$ as a measure of similarity between the decodable concepts of brain voxels across participants. Our objective is to apply a clustering algorithm using this metric in order to find areas in the brains of multiple participants that have a high conceptual similarity. We refer to these underlying concepts as shared decodable concepts (SDCs). We interpret the SDCs by retrieving a set of representative images associated with the brain-decoded CLIP vectors that are closest to the centroid of each SDC cluster. A schematic of the algorithm we apply is given in Figure 10.

## 3.1 CROSS PARTICIPANT CLUSTERING

Much of the prior work reporting semantically interpretable information in NSD focuses on modeling individual participants (e.g. Luo et al. (2023a;b)). This is because current methods that combine fMRI across participants without anatomical alignment, so-called *functional alignment* methods, require that participants see the same stimuli in the same order (e.g. Shared Response Model (Chen et al., 2015), Hyperalignment (Haxby et al., 2020)). However, in NSD image order is randomized and most images are seen only by a single participant, precluding the application of common functional alignment methods and other methods that require the same stimuli in each sample space (e.g., Canonical Correlation Analysis, CCA). Our approach leverages participant specific models that cast each voxel into CLIP's feature space, allowing us to use data that (a) aren't anatomically aligned, and (b) don't share exactly the same stimuli sets.

To discover shared brain-decodable concepts we apply a novel clustering method to the per-voxel model parameter vectors $w_i^{(k)}$ across all participants. We base our clustering method on the Density-Based Spatial Clustering of Applications with Noise (DBSCAN) (Ester et al., 1996) algorithm, which is parameterized by a neighborhood size $\varepsilon$ and a point threshold `minPts`. The algorithm can be summarized using the following steps: (1) Points that have at least `minPts` points in their $\varepsilon$-neighborhood are marked as *core* points; (2) construct a graph $\mathcal{G}$ where the vertices are core points, and there are edges between core points that have a distance less than $\varepsilon$; (3) clusters are formed by finding the connected components of $\mathcal{G}$ and (4) the remaining *non-core* points are added to clusters if they are within the $\varepsilon$-neighborhood of a core point, otherwise, they are marked as outliers not belonging to any cluster.

In our modified cross-participant DBSCAN, we redefine the core point threshold `minPts` as `minNeighbors`. Then, in step (1) a point $w_i^{(k)}$ is a core point if there are points from at least `minNeighbors` other participants within its $\varepsilon$ neighborhood. With this modification, a core point now identifies voxels that represent a brain-decodable concept that is shared with at least `minNeighbors` participants. Since there are 8 participants in the study, the maximum value of `minNeighbors` is 7. Steps (2) to (4) proceed as above.

Our second modification is to apply a within-participant expansion of clusters. We define a new hyperparameter $\varepsilon_{\text{expansion}}$ and apply a final expansion step: (5) All points inside the $\varepsilon_{\text{expansion}}$ neigh-

borhood of a point in a cluster become members of that cluster with the constraint that they must be from the *same* participant. The addition of this step was motivated by our early experiments, where we noticed that there were sometimes only one or two voxels belonging to certain participants for each cluster. This allows the cluster boundaries to grow slightly larger for each participant, and we found that this did not significantly change the cluster centroid or semantic interpretations. We set the expansion neighborhood size $\varepsilon_{\text{expansion}} = \min(\varepsilon + 0.05, 0.65)$ so that the cluster boundaries can grow slightly larger than the baseline neighborhood size $\varepsilon$, but not exceeding 0.65 as we noticed neighborhoods become over-connected as $\varepsilon$ approaches 0.7. Figure 1 outlines the application of our modified DBSCAN algorithm to modeling fMRI data.

We present results using a fixed value for `minNeighbors = 3`. This means that each cluster must include at least 4 out of 8 participants in the study. This allows for the discovery of SDCs that may show up less reliably in some participants. Inspired by the hierarchical version of DBSCAN (HDBSCAN) Campello et al. (2013), we ran our modified DBSCAN algorithm over a grid of $\varepsilon$ values, ranging from 0.8 to 0.4 in steps of 0.001. The resulting clusters are then organized into a hierarchical structure as defined by HDBSCAN. The first cluster considered contains all points at the highest value of $\varepsilon$. As $\varepsilon$ decreases, clusters may split into smaller subclusters. When a cluster splits, we classify its child(ren) clusters as follows: If two or more child clusters contain at least `minClusterSize` core points, they are added to the hierarchy as distinct subclusters. If only one child cluster meets this threshold, it is treated as a continuation of the parent cluster, while the remaining smaller clusters (with fewer than `minClusterSize` core points) are considered outliers. We show a selection of clusters from the hierarchy in Section 4 and the appendix, while the full hierarchy of clusters can be explored in our interactive viewer tool (Section A.13).

To reduce the impact of random initialization and combat the noise inherent in fMRI, we re-trained the decoder model 50 times for every participant and compute the average of the resulting parameter vectors. In other words, each averaged parameter vector $\boldsymbol{w}_i^{(k)}$ used in our analysis represents the average of 50 concepts that a voxel could represent depending on the random initialization of the decoder. We found that this significantly improved the quality and consistency of the clusters revealed by our modified DBSCAN algorithm.

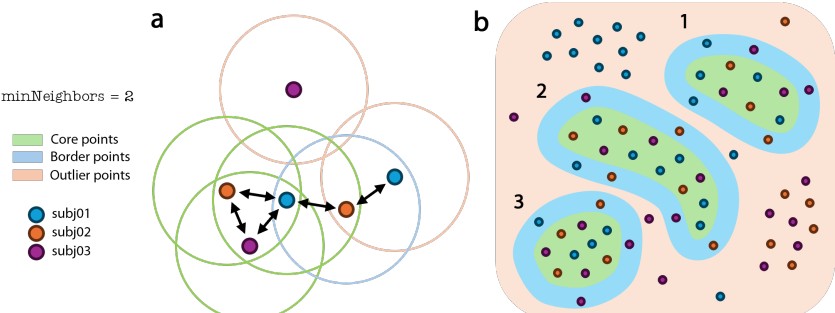

Figure 1: **Illustration of our DBSCAN variant applied to multi-participant fMRI data.** There are 3 participants in this example and `minNieghbors = 2`. **(a):** A zoomed-in view of a cluster with three core points. The outer ring around each point shows its $\varepsilon$-neighborhood and whether it is a core, border, or outlier point. Black arrows emphasize points that are neighbors. The points with neighboring points from at least 2 other distinct participants are marked as core points. Non-core points that neighbor core points are added to the cluster as border points. The remaining points are marked as outliers. **(b):** A zoomed-out sketch of a set of points that form 3 clusters. Since `minNieghbors = 2`, a high-density region will form clusters if and only if it contains points from at least 3 participants.

## 3.2 SELECTING REPRESENTATIVE IMAGES AND WORDS

To interpret the semantic meaning of an SDC cluster, we first computed the cluster centroid by taking the mean of all parameter vectors within the cluster $\bar{\boldsymbol{w}} = \frac{1}{|I|} \sum_{(i,k)\in I} \boldsymbol{w}_i^{(k)}$ where $I$ is a set of tuples identifying the voxel and subject indices for an SDC cluster. When the cluster voxels are above or

below baseline activation, the CLIP dimension represented by $\bar{w}$ is increased or decreased in the brain-decoded embedding, respectively. We note that DBSCAN has the ability to create non-globular clusters for which the centroid might not be a good representative. In our experience the centroid is within $\varepsilon$ of a core point within the cluster, implying that it is likely fairly representative of the cluster.

In order to find the representative images for an SDC cluster, we can consider the nearby image-embeddings to the cluster centroid $\bar{w}$. Instead of using the original CLIP embeddings for each image, we instead used the subject-specific *brain-decoded* embeddings $\hat{Y}_{\text{Test}}^{(k)}$. This allows us to focus specifically on the information that can be decoded from fMRI, which could be a subset of the information represented by CLIP space. For images that were viewed more than once we average the embeddings within and across participants. These averaged brain-decoded embeddings from held-out test data are denoted by $\hat{Y}_{\text{Test}}^{\text{AVG}}$. To select nearby images we define $D(w, Y) = \{d(w, y) | y \in Y\}$ which is the set of cosine distances between a CLIP vector $w$ to a set of CLIP embeddings $Y$. We take the cosine distances between the SDC cluster centroid $D(\bar{w}, \hat{Y}_{\text{Test}}^{\text{AVG}})$ and the *negated* centroid $D(-\bar{w}, \hat{Y}_{\text{Test}}^{\text{AVG}})$ and retrieve the images corresponding to the smallest values in these sets as the positive and negative representative images respectively.

Similar to selecting representative images, we can find sets of representative words for the SDC clusters. To accomplish this we first embed all 5 captions for each of the 73000 images in the NSD stimulus set to obtain $Y_{\text{text}} \in \mathbb{R}^{365000 \times 512}$. Unlike the representative images, these embeddings are not brain-decoded and are generated solely by the CLIP text encoder. To select the best representative captions we compute $D(\bar{w}, Y_{\text{text}})$ and $D(-\bar{w}, Y_{\text{text}})$ and select the captions corresponding to the 50 smallest distances in each set as the positive and negative captions respectively. We then utilize the wordcloud python package to create visualizations of the most frequently occuring words in the captions, which we present in appendices (Figure 20).

## 4 RESULTS

The clustering method described in chapter 3 not only identifies previously discovered functional areas but additionally identifies two new ones, both of which we explore in this section. We emphasize that there is no constraint that voxels within an SDC cluster be spatially adjacent in the brain, yet during visualization we consistently find contiguous patches both within and across participants. We can localize these patches of voxels to specific regions of interest (ROIs) both functionally and anatomically (See Section A.4 for further details).

### 4.1 FACES

One of the first reported functionally localized areas for higher-order vision was the fusiform face area (FFA) (Kanwisher et al., 1997). Our method identifies a face-related concept (Figure 2, cluster 88) that is localized to FFA and includes voxels from all 8 participants. This cluster also has some voxels in the extrastriate body area (EBA), likely because many face images include part or all of a person's body. We note that the positive representative images are not exclusively human and include a range of animal faces. The images often depict people eating and handling food, or holding other objects such as toothbrushes or cell phones.

Interestingly, the negative representative images often display bodies, but there is a striking *lack* of clearly visible faces. There are several examples where people are visible, but their faces are obscured or they are facing away from the camera. This suggests that there may be a dip in FFA activity for images where a person's face might be expected (and thus FFA is primed to activate) but not clearly visible.

### 4.2 FOOD AND COLOR

Previous work has specifically noted the correlation of very colorful images with food-related images Khosla et al. (2022); Pennock et al. (2023), and attempted to define food areas in the absence of color (Jain et al., 2023). Our method also identifies a large possibly food-related cluster that spans FFA, PPA, and V4 (cluster 114, Figure 3). At first inspection most of the positive representative images are food-related. However, we also observed many vibrant and colorful positive images that contained no food. Strikingly, the negative representative images are *entirely gray-scale*, providing

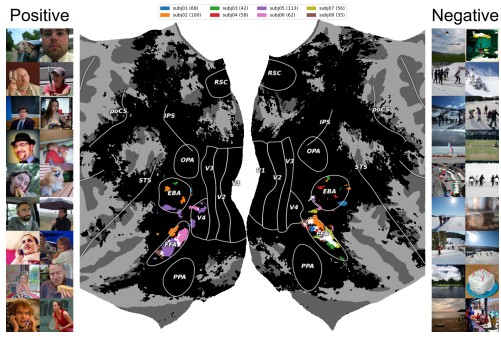

Figure 2: Cluster 88 ($\varepsilon = 0.593$). Positive images are strongly associated with faces, while the negative images represent depictions of people whose faces are not visible. Voxel clusters are primarily found in bilateral FFA and EBA.

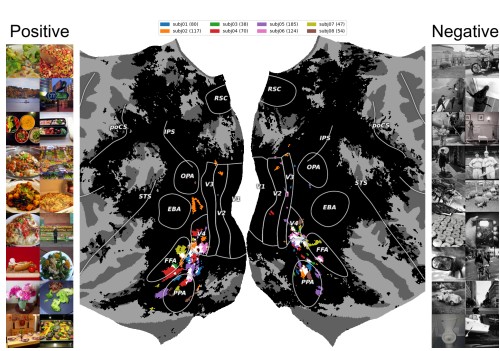

Figure 3: Cluster 114 ($\varepsilon = 0.555$). Positive images are associated with food and color. Negative images are entirely grayscale. Voxel clusters span an area between bilateral FFA, V4, and PPA.

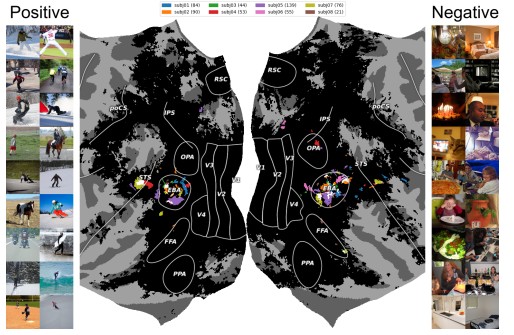

Figure 4: Cluster 106 ($\varepsilon = 0.569$). Positive images are strongly associated with presence of legs, while negative images are typically people at tables whose legs are obscured. Voxel clusters are primarily in bilateral EBA.

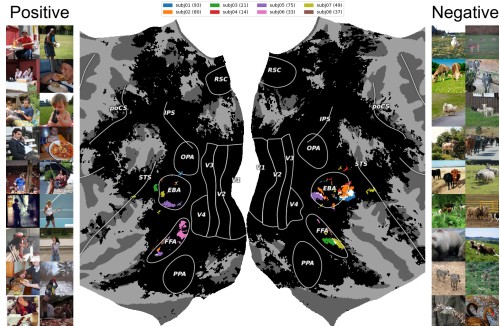

Figure 5: Cluster 89 ($\varepsilon = 0.593$). Positive images are strongly associated with hands and hand motion, while negative images are associated with animals (no hands). Voxel clusters are primarily in bilateral EBA and FFA.

very strong evidence that this cluster may be related to color. Thus, we speculate that this cluster is not specifically related to the identification of food, but might rather correspond to any colorful image. We discuss two other food-related clusters in supplementary section A.9.

## 4.3 BODIES

Our method identifies five notable body-related areas in and around EBA (Downing et al., 2001). The positive representative images for cluster 106 (Figure 4) show people and animals outside with an emphasis on legs and active movement. The negative images typically depict people indoors sitting with their legs obscured by tables. Cluster 89 (Figure 5) has a similar emphasis on hands instead of legs. People are displayed in a variety of contexts with their hands clearly visible, while the negative images are exclusively non-primate animals without hands. This cluster also shows indoor/outdoor contrast in the representative image groups and so there is some PPA activation. In Section A.8 we outline further body-related shared voxel clusters associated with motion and a distinction between single individuals and groups of people in a sub-region of EBA.

## 4.4 HORIZONTAL/VERTICAL CLUSTER

Very early work on the visual system discovered the tuning of the early visual system for lines of a particular orientation. Two clusters emerge that reflect this tuning. Cluster 60 (Figure 6) has a strong horizontal component with strong horizon lines or large objects spanning the middle of the visual

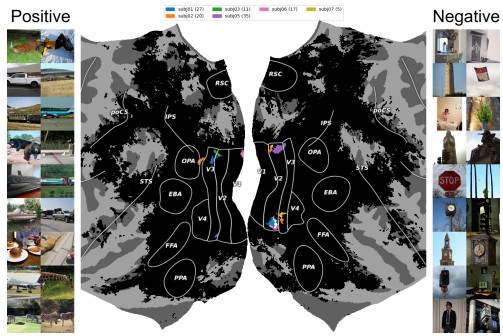

Figure 6: Cluster 60 ($\varepsilon = 0.668$). Positive images are associated with a horizontal mid-line, while negative images display a vertical mid-line. Voxel clusters are primarily in right hemisphere V2.

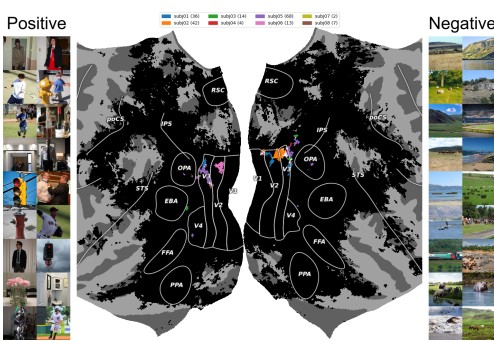

Figure 7: Cluster 65 ($\varepsilon = 0.657$). Positive images are associated with a vertical mid-line, while negative images display a horizontal mid-line. Voxel clusters are primarily in bilateral V3.

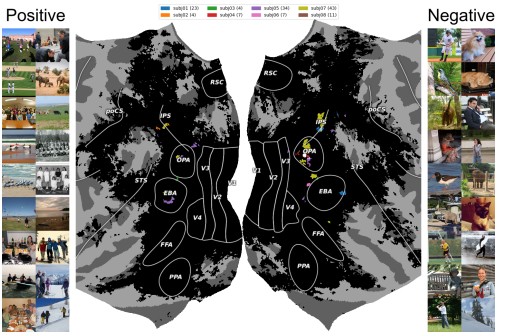

Figure 8: Cluster 111 ($\varepsilon = 0.563$). Positive images are associated with repeated objects, people, or other animals, while negative images are single objects or individuals. Voxel clusters are primarily located in OPA and IPS.

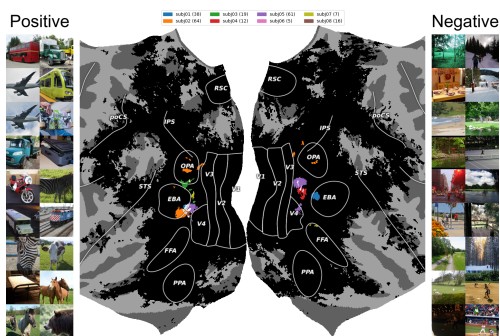

Figure 9: Cluster 67 ($\varepsilon = 0.654$). Positive images display close-ups of large animals or objects. Negative image show objects at a distance. Voxel clusters are located between EBA and V4.

field creating a horizon-like line. Notably, the negative images are images with a strong vertical component. The inverse is true for cluster 65 (Figure 7) which shows strong verticality in the positive images, and horizontal in the negative. Interestingly some of the negative images from cluster 60 appear as positive images in cluster 65, and vice versa. The localization of both clusters is primarily in the periphery regions of V2/V3. We hypothesize that the lack of foveal responses is related to the fact that horizontal and vertical paths, which define these orientation responses, likely cross in the foveal region. Therefore, peripheral responses are expected to be much more informative in distinguishing the identified horizontal/vertical components of these images.

## 4.5   REPEATED ELEMENTS (NUMEROSITY) CLUSTER

We observed that cluster 111 (Figure 8) has positively associated images typically with repeated elements of related objects, people, or other animals in similar orientations, while negative images typically contained singular instances of objects or individuals. We suggest that this SDC could be related to quantity processing and numerosity. We observed voxel clusters primarily in right OPA and IPS regions. As with other clusters, naturalistic images often implicates functional areas associated with place (OPA). The IPS has been widely reported in prior work to be associated with quantity processing, e.g. grammatical number processing (Carreiras et al., 2010), mathematical processing deficits (Ganor-Stern et al., 2020) and numerical processing (Dehaene et al., 2003; Dehaene, 2007; Nieder and Dehaene, 2009; Koch et al., 2023)). Our observations in the visual domain provide converging evidence that the right IPS is associated with quantity processing across multiple modalities.

### 4.6 Close vs Far Cluster

Cluster 67 (Figure 9) seems to represent big things (airplanes, busses) and close up pictures of larger animals (zebras, elephants). The negatively associated images are scenes that usually depict significant depth, including paths leading into the hills with animals or people in the distance. Konkle and Oliva (2012) and Sarch et al. (2023) explored the representation of image depth in cortex and also found that similar areas of cortex respond strongly to images with objects very close to the camera. In addition, Luo et al. (2023b) reported that this general area is sensitive to relatively large objects.

## 5 Conclusion

We proposed a novel mechanism to identify shared decodable concept clusters in a large multi-participant fMRI dataset of natural image viewing (Allen et al., 2022). To achieve this, we applied a contrastive learning approach using multimodal embeddings from CLIP (Radford et al., 2021) and a novel adaptation of the DBSCAN clustering algorithm (Ester et al., 1996). Our method identifies previously reported functional areas that code for various visuo-semantic categories, as well as identifying *novel* shared categories. Furthermore, our method exposes interpretability by identifying the most / least associated images with CLIP-based cluster centroids. In the case of faces, we find a set of positive / negative images where the negative images depict scenarios where a face might be expected, but is not visible. This suggests that the brain responses for this SDC cluster are organized in a way that specifically codes for the presence of faces in expected visual depictions. Food is a concept that is highly confounded with other visual criteria such as shape and color. By providing the most negatively associated images for a SDC cluster, we enabled easy interpretation of semantic coding, e.g. in the case where the negated images to this SDC cluster are all grayscale images (see Section 4.2). We further identify SDC clusters specific to bodies in motion, in groups, as well as hands and legs. We identify SDC clusters related to repeated elements (numerosity), sensitivity to indoor / outdoor scenes, cardinal orientation of objects in images (horizontal / vertical) and gradations of shape information (soft vs hard) and visual perspective (close vs far). Our method showcases how various techniques from machine learning can be employed to better characterize visual representations of semantic selectivity in the human brain.

### 5.1 Broader Impacts

Methods that decode neural activity could have substantial impacts on both neuroscience and broader society. Identifying new category-selective brain regions could assist in diagnosing neurological and psychiatric conditions, surgical planning, development of brain-computer interfaces, and potentially helping individuals with locked-in syndrome or related disabilities to better communicate.

However, there are also significant ethical considerations with technologies that decode neural responses. If these techniques are used outside controlled research environments, there is a risk that they could be used for intrusive monitoring of mental states. As with all emerging technologies, responsible deployment is needed in order for society as a whole to maximize benefit while mitigating risks.

### 5.2 Limitations and Future Work

While our approach offers many advantages, it is not without drawbacks. Firstly, we are limited by the stimulus images that were chosen for the NSD experiment. Although the use of a contrastive loss function helps mitigate this, our model may still be biased toward over-represented categories in the stimulus set. Additionally, the use of CLIP may introduce its own biases, as some signal in the brain responses may not be fully captured by the CLIP embeddings. The use of fMRI also presents certain limitations. Despite its high spatial resolution, its limited temporal resolution restricts the ability to capture rapid neural activity that may play a crucial role in visual processing. Future work could address these limitations by utilizing larger and more diverse fMRI datasets, incorporating higher temporal imaging techniques such as electroencephalogram (EEG) or electrocorticography (ECoG), and exploring alternative stimulus representations beyond CLIP.

Additionally, our clustering approach might merge concepts within regions of CLIP space that have relatively uniform densities, potentially missing interesting SDCs. This issue could be addressed

by adapting the hierarchical version of DBSCAN, or by using different criteria to merge the cross-participant core points into clusters. The use of hierarchical DBSCAN could consider a range of epsilons in the same clustering run, allowing for both more and less dense clusters to be identified. This may lead to both distributed and localized clusters emerging as the hierarchical clustering proceeds.

Furthermore, we acknowledge that our method is not a replacement for traditional hypothesis-driven experiments. It will be important for future work to test for alternative hypotheses for what might be driving responses in SDC clusters. Another interesting research direction is the application of this method to other sensory modalities beyond vision, such as auditory or somatosensory stimuli.

## 6 ACKNOWLEDGMENTS

This research was supported by the Natural Sciences and Engineering Research Council of Canada (NSERC), the Social Sciences and Humanities Research Council (SSHRC), the Digital Research Alliance of Canada (alliancecan.ca), New Frontiers in Research Fund (Exploration), and the Canadian Institute for Advanced Research (CIFAR). Compute resources were graciously supplied by Prairies DRI and the Digital Research Alliance of Canada. Alona Fyshe holds a Canada CIFAR AI Chair.

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

# A    SUPPLEMENTARY MATERIAL

## A.1    DERIVING SDC CLUSTERS

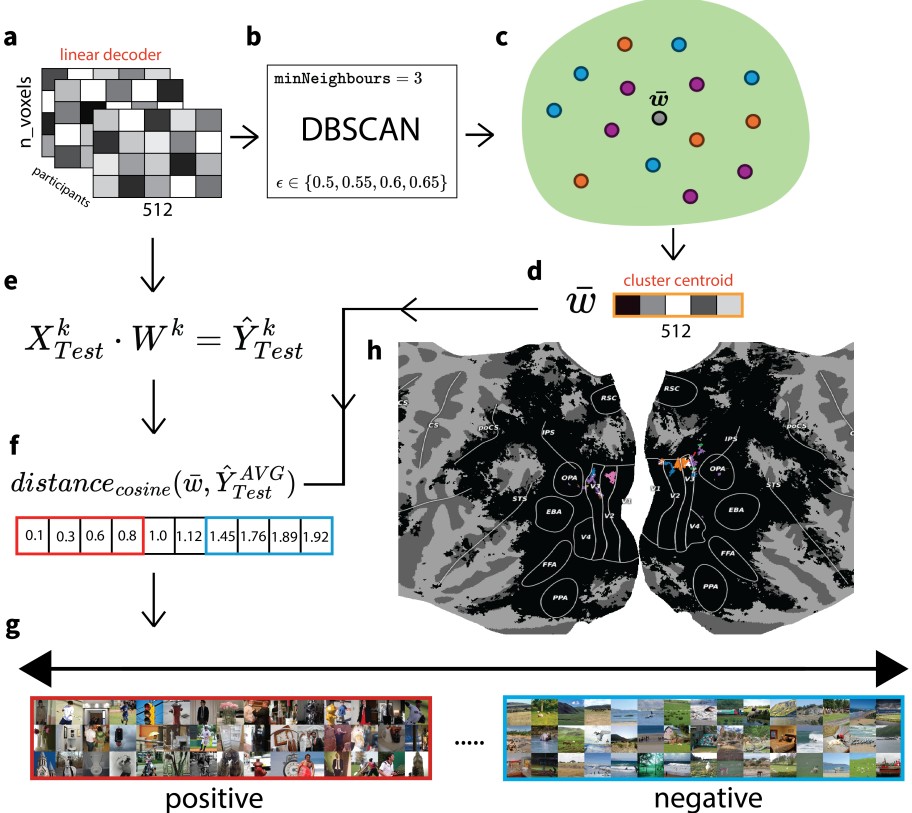

Figure 10: **Deriving SDC clusters. (a):** The participant-specific linear decoders derived in Figure A.2. **(b):** Our modified DBSCAN clustering procedure is applied to the linear decoders (See Figure 1 for details). **(c):** The DBSCAN algorithm discovers cluster centroids in high-dimensional space. **(d):** The 512-dimensional representations from the previous step are averaged over voxels and participants to derive a cluster centroid for each cluster derived from DBSCAN. We visualize the cluster centroid for the first DBSCAN cluster. **(e):** The linear decoders from **(a)** are applied to held-out fMRI data per-participant. Each participant saw images multiple times, so the matrix of predicted CLIP embeddings $\hat{Y}_{Test}^{k}$ is averaged over these repetitions and all linear decoding matrices are stacked (across participants) to give $\hat{Y}_{Test}^{AVG}$. **(f):** Cosine distance is calculated between the cluster centroids (e) and the brain-derived CLIP embeddings (f). **(g):** The images most associated with the cluster centroids (positive images) and most negatively associated with the cluster centroids (negative images) are identified. Positive / negative images for the SDC cluster pictured here appears to correspond to global vertical/horizontal orientation in the associated images. **(h):** Color-coded participant-specific voxel clusters are displayed on a flatmap of the brain's cortical surface in common *fsaverage* space (overlapping areas are displayed in white). Regions of interest labels are highlighted on the flatmap image in white outlines. For the specified cluster (d), whose positive / negative images are associated with orientation, the flatmap indicates bilateral shared voxel clusters in early visual cortex.

## A.2    DECODING CLIP REPRESENTATIONS

The decoding model is trained for 5000 iterations (29 to 45 epochs depending on the participant's training set size) with the Adam optimizer, a batch size of 128, and a fixed learning rate of $1e^{-4}$. Data augmentation is applied to help slow overfitting by adding random noise to training samples

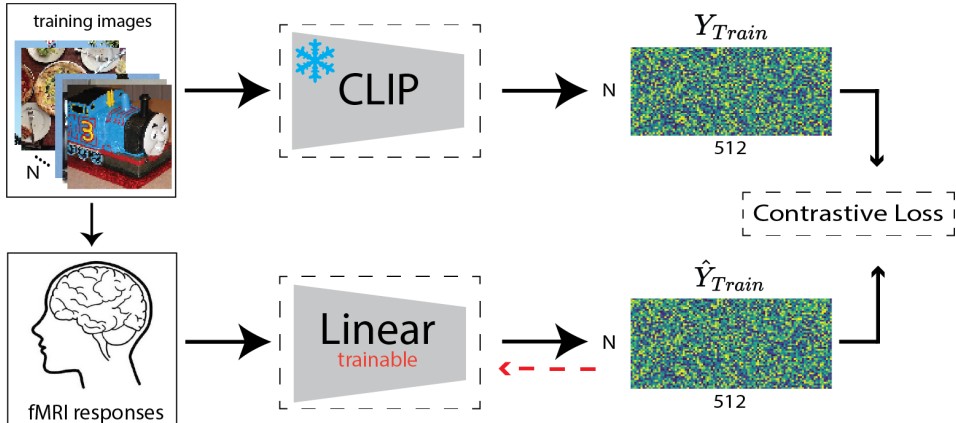

Figure 11: Decoding CLIP representations from the brain. We create CLIP representations by passing NSD training set stimuli to CLIP (frozen). The NSD team showed these same images to human participants during fMRI scans. We train a linear decoder with contrastive loss (Eq. 2) to predict CLIP embeddings from the fMRI responses to the corresponding images. Black arrows represent the flow of data through the procedure and dashed red lines represent gradient updates used to train the model.

$x_i \leftarrow x_i + z$ sampled from a normal distribution $z \sim \mathcal{N}(0, \sigma^2)$ where the noise standard deviation $\sigma$ is a hyper-parameter. We set $\tau = 0.03$ and $\sigma = 0.1$ in our implementation. Hyper-parameters were selected based on performance on the validation set. The exact number of voxels input to the model was $[17883, 18358, 13476, 11899, 17693, 18692, 9608, 6772]$ for subjects 1 through 8. We compare our contrastive decoder to a baseline ridge regression model trained on the same data. We used grid search to select the best ridge regularization parameter $\lambda \in \{0.1, 1, 10, 100, 1000, 10000, 100000\}$ using the validation data. The optimal $\lambda$ was 10000 for participants 1, 2, 5, 6, and 1000 for participants 3, 4, 7, 8.

## A.3 COMPUTE RESOURCES

The decoder models were trained on an NVIDIA GeForce RTX 2060. Training time was approximately 2 minutes for each model. The total training time for 50 models for each of the 8 NSD participants was about 13 hours. The DBSCAN clustering algorithm uses minimal CPU resources and executes within a few minutes.

## A.4 REGION OF INTEREST IDENTIFICATION

In order to localize shared voxel clusters in our visualizations, we overlaid region of interest boundaries onto the flatmap visualizations in Freesurfer's *fsaverage* space. For functional ROIs, we retraced the ROIs given in the NSD dataset during the functional localization experiments (fLoc, Stigliani et al. (2015)).

## A.5 TOP-K ACCURACY FOR CLIP DECODING

See Figure 12.

## A.6 VOXEL NORMALIZATION

Brain responses were per-voxel normalized to have zero mean and unit standard deviation within each scanning session. As a result, we allow brain response values to be positive or negative. In order to study both positive and negative aspects of stimuli-driven brain responses, this is a

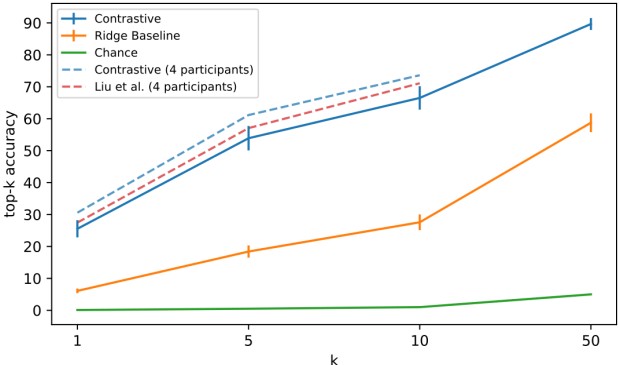

Figure 12: Top-k accuracy for CLIP decoding using the proposed contrastive decoder and a ridge regression baseline. The contrastive decoder implementation outperforms the ridge regression baseline across various values of $k$. Chance performance is given in green. Accuracy calculated on held-out data. Bars indicate the standard error (SE) across the 8 NSD participants. Additionally, we include a comparison to Liu et al. (2023) who report the mean top-k accuracy across 4 subjects (1, 2, 5, 7) using brain responses to 982 test images (dashed red line). We average the results of our own contrastive decoder across these participants for a fair comparison (dashed blue line). Despite a slightly larger test set, our contrastive decoder has a higher accuracy for all reported values of k.

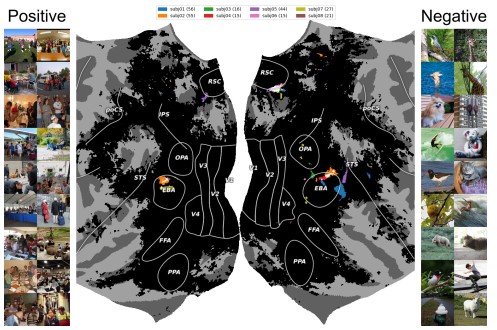

Figure 13: Cluster 110 ($\varepsilon = 0.563$). Positive images display crowds, while negative images are of individuals. Voxel clusters are in bilateral EBA.

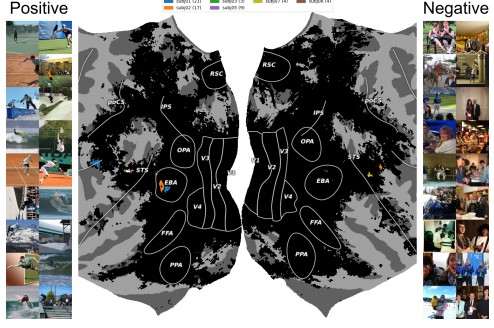

Figure 14: Cluster 107 ($\varepsilon = 0.569$). Positive images show people jumping or leaping, while negative images are of people standing still or sitting. Voxel clusters are in and around left hemisphere EBA.

necessary preprocessing step as the beta weights in NSD are calculated with respect to a baseline gray background which is not zero-centered (Allen et al., 2022).

## A.7 PLACES

Cluster 115 (Figure 17) strongly overlaps with PPA and displays outdoor scenes with consistent vegetation for the positive images. The negative images show indoor scenes where human-made objects are prominent, with a distinct lack of vegetation.

Cluster 79 (Figure 21) displays indoor scenes for the positive representative images. This cluster is mostly localized in OPA with some voxels in PPA. The positive images depict cluttered indoor scenes such as kitchens, work spaces, and living rooms. There is typically a flat surface such a desk, countertop, or table at the focal point of the image. The negative images depict outdoor scenes with animals.

Cluster 122 (Figure 22) overlaps strongly with PPA, OPA, and RSC. The positive representative images usually display an urban area where the camera is looking forward down a long path or road. In contrast, the negative images contain many close-up images of objects on tables where

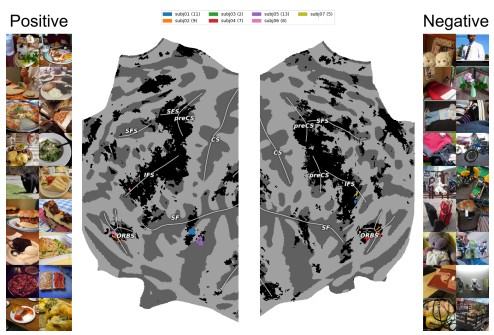

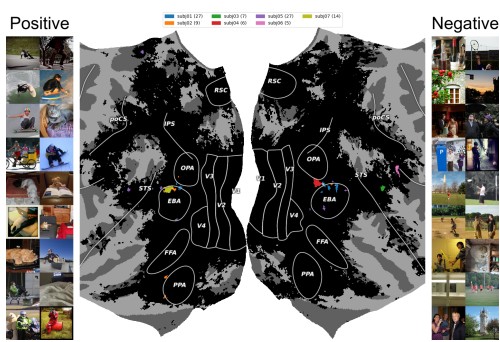

Figure 15: Cluster 64 ($\varepsilon = 0.657$). Food-related images with shared voxel clusters in anterior regions of the brain.

Figure 16: Cluster 87 ($\varepsilon = 0.596$). Positive images show people with bent legs or animals laying down. Negative images primarily show people standing. Voxel clusters are in bilateral EBA.

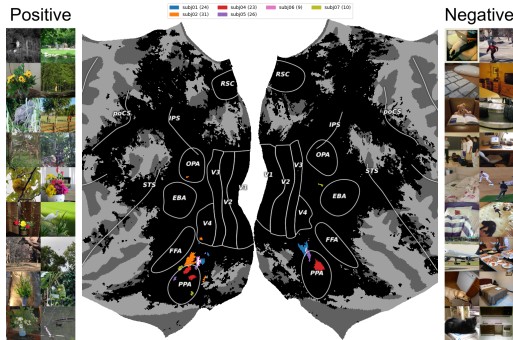

Figure 17: Cluster 115 ($\varepsilon = 0.555$). Positive images display scenes with plants and foliage, while negative images show human-made objects. Voxel clusters are in bilateral PPA.

the surrounding environment is not visible. This suggests that this cluster may be related to scene geometry, navigation, or the reachability of locations in a scene (Epstein and Kanwisher, 1998; Dilks et al., 2013; Maguire, 2001).

## A.8 BODIES (CONTINUED)

In Section 4.3 we identified various SDCs relating to the presence of bodies in images. We expand on a few further observations here. Cluster 110 (Figure 13) shows groups of three or more people in the positive images, with a strong focus on an individual person, animal, or object in the negative images. Cluster 107 (Figure 14) appears to be related to full-body leaping motions, with the negative images showing people sitting or standing still. In contrast, cluster 87 shows a mixture of images of people who are crouched or sitting, along with images of pet cats and dogs laying down.

## A.9 FOOD

Cluster 64 (Figure 15) is shared across seven participants and is localized to frontal brain regions. Voxels are located in the orbital sulci, as well as the boundary between the triangular part of the inferior frontal gyrus and the inferior frontal sulcus. We noted that the positive images to contain images of food on plates, as well as a few non-food images (bear, skiing, playing baseball). We had trouble discerning a strong core concept in the negatively associated images, aside from a distinct lack of food.

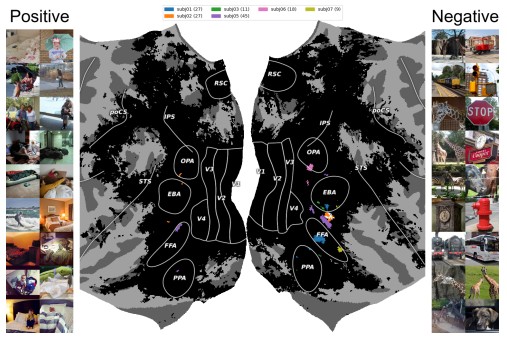
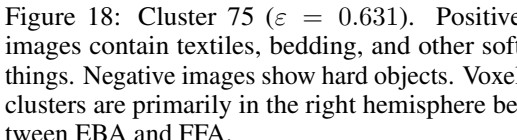
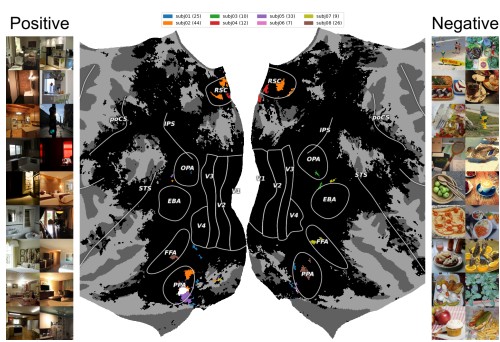

Figure 18: Cluster 75 ($\varepsilon = 0.631$). Positive images contain textiles, bedding, and other soft things. Negative images show hard objects. Voxel clusters are primarily in the right hemisphere between EBA and FFA.

Figure 19: Cluster 119 ($\varepsilon = 0.534$). Positive images display a contrast between a light source and a dark environment. Negative images show uniform ambient lighting. Voxel clusters are primarily in bilateral RSC and PPA.

## A.10 SOFT VS HARD CLUSTER

Cluster 75 (Figure 18) shows images with a strong focus on clothing, bedding, and other textiles. The negative images show trains, and concrete-dominant architecture. The voxels in this cluster are mostly found in the right hemisphere bordering the area between FFA and EBA.

## A.11 LIGHTING-RELATED CLUSTER

Cluster 119 (Figure 19) includes voxels in RSC and PPA. The positive images depict scenes with a high contrast in lighting. There is typically a dark environment with a bright light or a window that is partially illuminating the room. Conversely, the negative images depict close-up pictures of objects with uniform ambient lighting. This strongly suggests that this cluster of voxels responds to images that display a high contrast in lighting.

## A.12 REPRESENTATIVE WORD CLOUDS

In Figure 20, we present word clouds containing the most frequent words in the representative captions for each image. In most cases the word clouds support our interpretation of each underlying cluster concept. For example the word cloud for the lighting cluster (Figure 19) containing the words lit, lamp, sun, illuminated, dark, dimly, brightly, all of which are indicative of lighting conditions. Similarly, the repeated elements cluster (Figure 8) is well-characterized by words like group, together, formation, squadron, herd, which emphasize the concept of repetition or grouping.

In some cases, while the word clouds provide partial support for our descriptions, the thematic patterns become more apparent when examining the representative images. For example the positive representative words for the hands cluster (Figure 5) contains action-oriented words like as cutting and slicing, and the legs cluster (Figure 4) features words such as riding and running. However it is difficult to identify the overall theme of hands or legs without the representative images.

The word clouds for the food clusters offer additional insights. The food and color cluster (Figure 3) is strongly indicative of favoring the color versus grayscale concept. In contrast, the anterior food cluster (Figure 15) has a strong association to food-related terms like food, eating, plate, avocado, and feeding.

## A.13 INTERACTIVE RESULTS VIEWER

Alongside this paper, we release an interactive web application that allows for the exploration of the full hierarchy of results. A preview of our viewer is displayed in figure 23. The viewer can be accessed at the following URL: `fyshelab.github.io/brain-viewer`

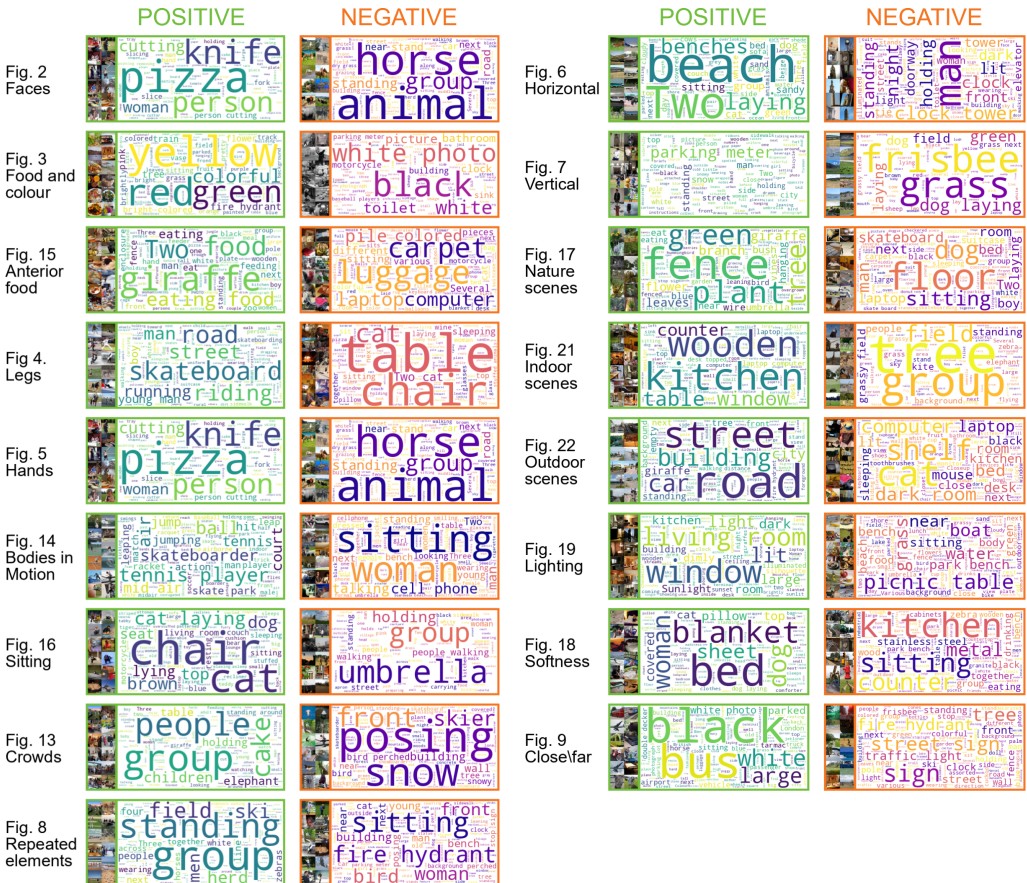

Figure 20: Word clouds for all SDC clusters. The positive and negative representative words are displayed side-by-side along with a subset of the representative images. The text to the left identifies the corresponding flatmap figure and our interpretation of the SDC.

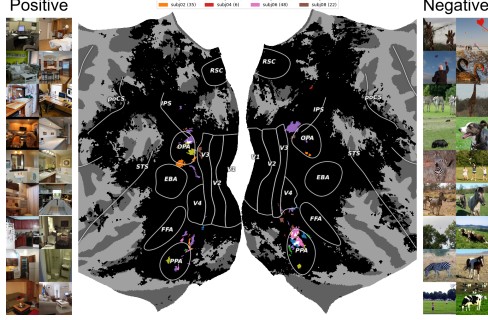

Figure 21: Cluster 79 ($\varepsilon = 0.607$). Positive images show indoor scenes with desks and tables. Negative images show outdoor scenes with animals. Voxel clusters are primarily in bilateral OPA and PPA.

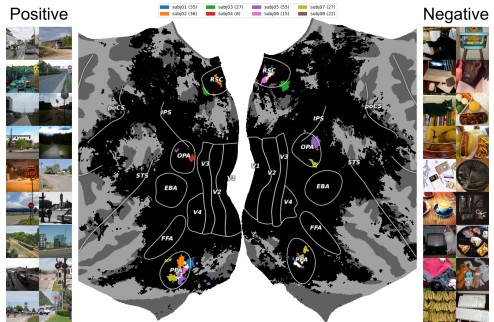

Figure 22: Cluster 122 ($\varepsilon = 0.528$). Positive images show scenes with large open areas. Negative images are close-ups of objects that have no scene geometry. Clusters span bilateral RSC, OPA, and PPA.

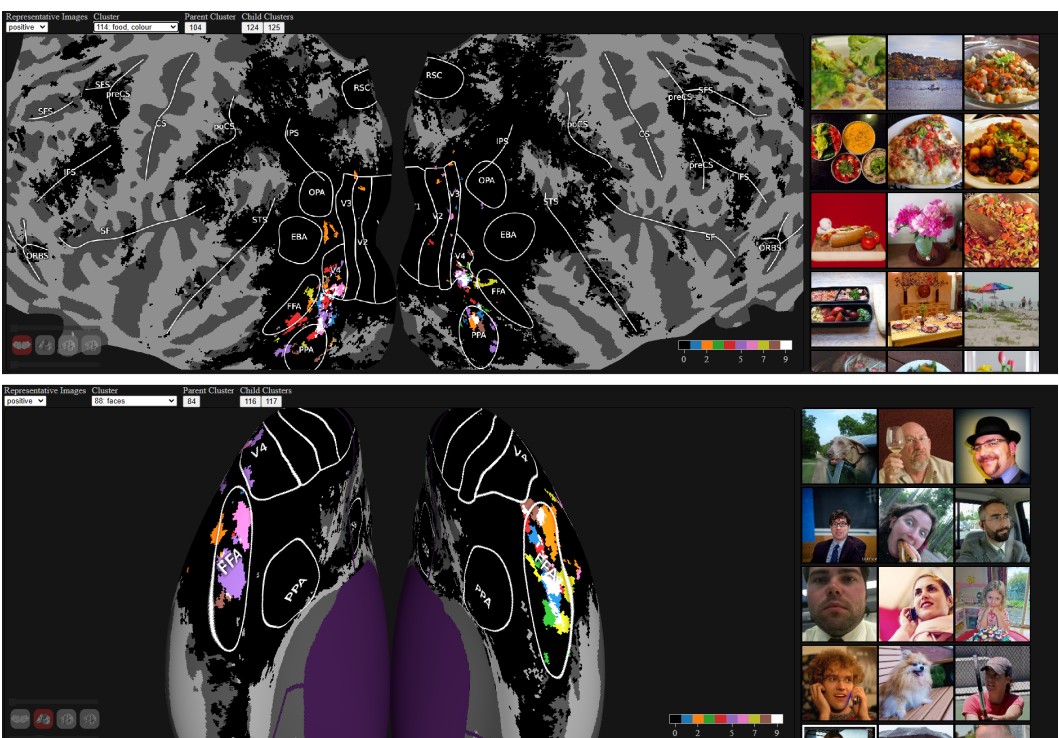

Figure 23: Preview of our interactive viewer application. The user can select `minNeighbors`, $\varepsilon$, whether to display the positive or negative representative images, and the cluster ID. The cortical surface can be toggled between flattened, inflated, pial, or white matter surface meshes with the controls in the bottom left corner.

## A.14 PROJECT CODE

Code to reproduce this work can be found at the following URL: `https://github.com/fyshelab/contrastive-decoding-iclr2025`

