# OpenReview forum: "Finding Shared Decodable Concepts and their Negations in the Brain"
_ICLR.cc/2025/Conference — ICLR 2025 Poster_

### Official Review · Reviewer_UQYg · 2024-11-02

**Soundness:** 3
**Presentation:** 3
**Contribution:** 3
**Rating:** 8
**Confidence:** 4

**Summary:**

This paper presents a novel technique for discovering and interpreting functionally specialized clusters in the brain. Using the Natural Scenes Dataset, the authors train linear decoders to map single voxel activities onto CLIP embedding vectors for the respective images seen in the neuroimaging experiment. Then, the authors leverage the similarity of voxel->CLIP prediction weights to perform clustering. Specifically, the authors develop a variant of the DBSCAN algorithm that leverages across-subject consistency in order to find the most consistent and semantically meaningful components. The authors present the results for several components, showing the spatial topography as well as the most strongly and weakly activating images. The authors interpret the components in light of both their spatial topography, and the content that appears present in the strong and weak activating images.

**Strengths:**

- Interesting method
- The use of a contrastive loss in the fitting of clip latents from voxel activities is particularly interesting
- The modification of DBSCAN is elegant, and provides an intriguing way of aggregating information across subjects
- Good figures

**Weaknesses:**

- The early vision components (horizontal and straight lines) activate parts of V1-V3 that represent the periphery (if the authors are unaware: on the flat map, the center of the strips of V1-V3 corresponds to foveal bias, and moving outward corresponds to increasing peripheral eccentricity). This isn't discussed but seems important.

- The results are very qualitative. The paper could be strengthened by making some objective quantifications of the content of different components, rather than just "reading the tea leaves".

- This paper joins with a serious of related papers exploring visual components (e.g. Khosla et. al 2022; BrainDIVE, Luo et. al 2023 NeurIPS; BrainSCUBA, Luo et. al 2024 ICLR), to name a few. I'm not sure if there is any single result here that is particularly novel. This makes the paper less compelling. Additionally, the method is not compared to other methods, so it is unclear whether it holds any advantages.

- The paper is very light on references. In my opinion, having 20 references for a paper on such a popular topic is a bit of a disservice to the field.

- This sentence is confusing: "We emphasize that there is no constraint that voxels within a shared visual semantic cluster be spatially adjacent in the brain, yet during visualization we consistently find contiguous patches both within and across participants". Please explain how this can be. The algorithm appears spatial, in that core points are defined as having at least minNeighbors points within their epsilon (spatial) neighborhood. Additionally, the epsilon expansion appears to be highly spatial in nature. If it is truly the case that there is no spatial constraint, it would be a mistake to simply call the algorithm a DBSCAN variant -- it would be better to say that it is inspired by DBSCAN, then give it a new name to avoid confusion. Actually, this is probably a good idea regardless.

**Questions:**

- Can you explain this sentence: "We emphasize that there is no constraint that voxels within a shared visual semantic cluster be spatially adjacent in the brain".

- How does this paper go above and beyond the findings of other recent works exploring functional selectivity using the Natural Scenes Dataset?


- Can you provide some intuition for why the contrastive fitting of voxel -> clip latents outperforms ridge regression?

---

> ### Author Response · Authors · 2024-11-20
> **Response UQYg**
>
> We thank the reviewer for kindly taking the time to provide useful feedback on our submission. We plan to split our response across the identified points. We hope this aids the reviewer in considering our response.
>
> Foveal bias
>
> We thank the reviewer for pointing out the observation of the potential bias away from the fovea in V1 responses. This is an interesting point. We discussed this phenomena during the write-up of our results and I can summarize our hypotheses. CLIP’s sensitivity relates to cohesive, caption-based information, which might be at odds with the orientation column sensitivity of V1 regions. The varying orientations seen in NSD images appear at varying points in images (e.g. clock towers are not always at the same point in images, images of beaches or airplanes are variably higher or lower) and this is detected more robustly as information is pooled together in slightly higher visual regions (V2/V3).  However, the spatial scale of the orientation columns in very early visual regions precludes extraction via our method because the CLIP dimensions don’t cluster around these low level visual features. This is why we believe there is a lack of decodable information in V1 using our method. As the reviewer describes, it certainly does seem potentially relevant and requires further investigation that would take us slightly out of scope in the current formulation of the paper.
>
> Qualitative focus
>
> We agree there is a level of human subjectivity involved in interpreting the clusters.  Our approach is a new paradigm for hypothesis generation and this is primarily where we see a benefit of our proposed approach. The potential confounds raised are valid, but until now a system like this (that contrasts two polarities of a concept vector) didn’t exist.  All data-driven analyses are likely to have similar issues with subjectivity.  We agree that future work to automate this analysis is interesting. Perhaps we could use the word clouds to facilitate such an analysis. In addition, the reviewer’s comments actually support the point we make: our method allows for the critical inspection of concept representations. The positive and negative images provide additional signals as to what concepts might actually represent, leading us towards more clear hypotheses for future testing.
>
> Novelty
>
> We will update the motivation of our approach particularly with regard to the other work done on NSD as we strongly believe our method has added the methodological toolkit. Our approach is the first one to make use of below-baseline responses to better constrain the potential meanings of the positive responses (e.g. In one cluster where food images being a primary positive cluster concept, we noted the negative images were all black-and-white, leading to our interpretation of the cluster as being color- rather than food-related). In addition, we also introduce a new clustering method that is able to find highly arbitrary cluster shapes (thanks to DBSCAN) reimagined to work with multi-subject data. In addition, we can utilize data when the images are viewed by single participants; we do not require that responses are to shared stimuli. These are novel aspects of analyses that are not currently seen in other approaches. We will update the paper to better reflect these points.
>
> Citations
>
> We appreciate the comment on the missing / limited citations. Numerous citations were genuinely omitted in error as we made our final edits to satisfy the page requirements and these were not then re-introduced into other parts of the paper. Thank you for bringing this to our attention. We have sought to identify the missing citations and have added them in, as well as adding in others to better contextualize the current work and prior work done on category selectivity in the brain.
>
> Sentence Clarification
>
> Our DBSCAN variant operates over distances defined in participants’ brain-decoded CLIP space, not the original anatomical space of brain data in which voxels or vertices have a spatial dimension. The distance in CLIP’s feature space is used for clustering, so voxel weight vectors (defined in CLIP space) can be clustered even if they are from voxels far apart in the brain. We don’t impose any (physical) constraint on voxel proximity during clustering. Nevertheless, we find that our concept clusters still largely contain spatially close voxels (in physical brain space). We apologize for the lack of clarity on this point and we have updated the paper to better reflect this.
>
> Contrastive loss versus ridge regression
>
> Like ridge regression, contrastive loss optimizes predictions to be close to the target.  But contrastive loss has an additional constraint that pushes predictions away from *other* target values.  This means that in the case that a target can’t be fit exactly, the model has additional information to at least make the prediction far from other target values which improves prediction accuracy beyond what ridge regression can do.

---

> > ### Comment · Reviewer_UQYg · 2024-11-25
> >
> > Thank you for your detailed response. While awaiting your revision, I will add some thoughts here.
> >
> > 1) (anti-)foveal bias: I agree with the content of your response, but it doesn't address why the component picks up on voxels corresponding to the periphery rather than the fovea.
> >
> > 2) citations: thank you for explaining this and vowing to address it.
> >
> > 3) sentence clarification: thank you for explaining this and I apologize for my prior confusion on this point. I now see that the distances are in terms of clip vectors derived from voxel encoding model weights. I now agree that the spatial contiguity of clusters is an interesting finding that is not guaranteed by the method. It would be helpful if the authors could comment on the degree of spatial smoothing used in their fMRI pre-processing pipeline.
> >
> > 4) contrastive loss vs. ridge regression: thanks for the explanation.
> >
> > 5) novelty and qualitative focus: I would strongly recommend the authors to make some substantial edits in their revision to make it clear exactly what new understanding this paper offers, above and beyond prior works.

---

> > > ### Author Response · Authors · 2024-11-26
> > > **Additional response to UQYg**
> > >
> > > Thanks for your follow up comment, and allowing us the chance to respond. Please see our comments below. We hope they help to clarify any remaining doubts and we recognise how addressing these points via additions to the paper (which we have just uploaded) will make for a more comprehensive contribution.
> > >
> > > 1. Lack of foveal bias - Because the horizons/vertical objects are not perfectly aligned in the center of the images, if we were to consider the pixels most indicative of the horizons/vertical objects they would likely be a thick horizontal/vertical band, which would actually overlap substantially in the foveal region.  The regions that would be most distinguishing for the horizons/vertical objects would actually be at the edges of the image.  This is our intuition about why there are no foveal areas included in the corresponding concept clusters. We will add a note about this to the paper.
> > >
> > > 3. We did not perform any additional spatial smoothing in our pipeline over and above what was applied in the release of the NSD data itself. In NSD the authors specify that they avoid any unnecessary spatial smoothing to maintain higher levels of information in the signal (“Overview of Data Processing” in NSD paper). The smoothing applied that the authors said was unavoidable related to a 3D FWHM Gaussian kernel (2mm), which is on the minimal end of the scale.
> > >
> > > 4. We will edit the paper to make the novelty more clear: we have introduced a new clustering method that utilizes a representational space to identify related voxels across participants.  Our method does not require spatial alignment of participants, but allows for the discovery of cross-participant relationships.  We also introduce the notion of least associated images which has proven quite telling for several of our clusters, sometimes enforcing the cluster meaning, and sometimes helping to refine the cluster meaning.
> > >
> > > Thank you for your additional comments.

---

### Official Review · Reviewer_LDPq · 2024-11-03

**Soundness:** 4
**Presentation:** 3
**Contribution:** 3
**Rating:** 8
**Confidence:** 3

**Summary:**

This paper proposes a new method to discover semantically interpretable fine-grained functional organization of fMRI responses that are consistent across participants. It then applies this method to the NSD data form Allen et al, reporting both established functional responses (FFA etc and responses claimed to be novel (e.g. for particular body parts, numerosity, etc).

I recommend acceptance of the paper because the method proposed is useful, the writeup is clear, and the analyses appear solid.

**Strengths:**

Strengths of the paper are that it tackles an important problem, applies it to a beautiful data set, and reports interpretable results that either serve to establish the validity of the method (via replication of established findings) or report intriguing new results. Another strength is the emphasis on the lowest responses of functional clusters, something that is widely appreciated in the field but rarely if ever explicitly discussed. A third strength is the care that was taken to re-calculate the per-voxel noise ceiling estimates on training data only, a small but important point.

**Weaknesses:**

The main weaknesses of the paper concern the failure to situate the paper in the context of relevant prior work. Some of the results are described as novel without reference to related findings published previously, For example, the claim of selectivity for different aspects of the body should acknowledge a large prior literature on this topic, e.g.:
Orlov, T., Makin, T. R., & Zohary, E. (2010). Topographic representation of the human body in the occipitotemporal cortex. Neuron, 68(3), 586-600.
Weiner, K. S., & Grill-Spector, K. (2011). Not one extrastriate body area: using anatomical landmarks, hMT+, and visual field maps to parcellate limb-selective activations in human lateral occipitotemporal cortex. Neuroimage, 56(4), 2183-2199.
Bracci, S., Caramazza, A., & Peelen, M. V. (2015). Representational similarity of body parts in human occipitotemporal cortex. Journal of Neuroscience, 35(38), 12977-12985.
Bracci, S., Ietswaart, M., Peelen, M. V., & Cavina-Pratesi, C. (2010). Dissociable neural responses to hands and non-hand body parts in human left extrastriate visual cortex. Journal of neurophysiology, 103(6), 3389-3397.
big-small close/far this is the ref:
Konkle, T., & Oliva, A. (2012). A real-world size organization of object responses in occipitotemporal cortex. Neuron, 74(6), 1114-1124.
And concerning the proposed method, data-driven analyses of fMRI data especially using clustering have been reported for at least 15 years, and this should be at least acknowledged, e.g.:
Golland, Y., Golland, P., Bentin, S., & Malach, R. (2008). Data-driven clustering reveals a fundamental subdivision of the human cortex into two global systems. Neuropsychologia, 46(2), 540-553.
Lashkari, D., Vul, E., Kanwisher, N., & Golland, P. (2010). Discovering structure in the space of fMRI selectivity profiles. Neuroimage, 50(3), 1085-1098.
Yeo, B. T., Krienen, F. M., Sepulcre, J., Sabuncu, M. R., Lashkari, D., Hollinshead, M., ... & Buckner, R. L. (2011). The organization of the human cerebral cortex estimated by intrinsic functional connectivity. Journal of neurophysiology.
and contrasted with alternative data-driven methods using ICA, NMF, etc, e.g.:
Norman-Haignere, S., Kanwisher, N. G., & McDermott, J. H. (2015). Distinct cortical pathways for music and speech revealed by hypothesis-free voxel decomposition. Neuron, 88(6), 1281-1296.
Khosla, M., Murty, N. A. R., & Kanwisher, N. (2022). A highly selective response to food in human visual cortex revealed by hypothesis-free voxel decomposition. Current Biology, 32(19), 4159-4171.

**Questions:**

Recommendations to improve the paper:

1. At least cite some of the relevant prior literature as discussed above.
2. It would be useful if the authors could clarify whether the "suppressed" responses are simply lower than those to other stimuli, or whether they are actually lower than a minimal baseline (e.g. the response when no stimulus is presented). In this reviewer's opinion "suppression" should refer to a response below the no-stimulus baseline, and it is not clear that is the case here.
3. The claim that food-related voxel clusters " span bilateral FFA, V4, and PPA." doesn't make sense given the usual definition of FFA and PPA as face and place selective. Presumably what is mean here is that food-related voxels are found near the FFA and PPA.
4. The authors should show how well the brain data reproduces CLIP embeddings (and how this changes over training). That is, show how similar Y_hat is to Y_CLIP.
5. How does using the CLIP dimension (w_i) improve on just using the neural response values (x_i) in this method?
6. Perhaps comment on whether these results tell us more about the brain or about CLIP.

---

> ### Author Response · Authors · 2024-11-22
> **Response LDPq**
>
> We thank the reviewer for kindly taking the time to provide useful feedback on our submission. we plan to split our response across the identified points. We hope this aids the reviewer in considering our response.
>
> Missing Citations (Q1)
>
> Thank you for pointing out some highly relevant citations. We had numerous missing citations due to a last-minute reorganization of the paper in order to fit the page requirements. This caused us to miss citations as they were not re-inserted into other suitable locations in the paper. We also appreciate the several new citations provided by our reviewer; they do better situate the work within a more descriptive prior context. The paper has been updated to reference this important prior work.
>
> Suppressed Responses (Q2)
>
> The fMRI data is preprocessed so that zero is equal to the response to a gray background.  However, during the training process, the data is mean-centered, which likely moves zero slightly away from response to a gray background. We are investigating this further, but are fairly confident that most negative responses would remain negative even after mean centering.
>
> Localization of Food Area (Q3)
>
> The reviewer is correct. We will reword this . Recent work on the food area largely implicates the boundary regions between FFA and PPA (Jain et al., 2023. Selectivity for food in human ventral visual cortex; Khosla et al. 2022. A highly selective response to food in human visual cortex revealed by hypothesis-free voxel decomposition). Our results find food and color related voxels around these areas in addition to some activity in V4, which we expect to drive the color component. The spatial distribution of our findings fits well with the broad localization of food components in prior work.
>
> Brain -> CLIP Quality (Q4)
>
> We’ve measured the Pearson correlation of our predictions and the per-subject averages are around $0.2$, meaning that we are predicting the CLIP embeddings with some accuracy. But, we believe that the top-K accuracy is the right way to measure performance of a contrastive model because the loss function explicitly optimizes for small distance to the target, but also large distance to the targets associated with other images.
>
> Neural Responses vs CLIP (Q5)
>
> Each participant viewed different images, so it would not be possible to take the distance between the voxel response vectors across participants in a clustering method. Training a decoder that predicts CLIP embeddings from brain data is what enables comparison of brain responses across participants even though different images were viewed.
>
> Results about the Brain or CLIP? (Q6)
>
> Our results tell us a bit about the brain and a bit about CLIP.  For instance, for the brain, the results tell us what is detectable at a finer-grained scale in terms of concepts, which is useful to generate novel hypotheses for future fMRI experiments.  Our work also indicates that there are some visual details that don’t show up in CLIP.  For example it appears that early V1 orientations do not show up in our results, suggesting that these exist at an idiosyncratic level not detectable in concepts found in CLIP.

---

> > ### Comment · Reviewer_LDPq · 2024-11-26
> > **Reply by Reviewer LDPq**
> >
> > Thank you for your detailed response and clarification. I look forward to seeing the final manuscript with the edits and updates you have mentioned. I’m particularly interested in seeing how mean centering the fMRI data impacts the negative responses in your results.
> >
> > Has the current manuscript been updated with all the citations, or will there be further changes in the camera ready version? After looking through the current pdf it seems that there are still some references that could be added.
> >
> > Overall, you have addressed the majority of our concerns and questions, and this reviewer had decided to keep their original score.

---

> > > ### Author Response · Authors · 2024-11-26
> > > **LDPq additional response**
> > >
> > > We received some further questions from a different reviewer and we have just finished addressing those and responding to those points. This meant we had not yet uploaded an amended version with the additional citations. We have just updated the PDF a moment ago and this version is our updated revision that takes into account comments and actionable points that you and the other reviewers have very kindly given us. We are extremely thankful for the kind, considerate and relevant points you have all raised and we hope we have been able to clarify any points where there might have been a lack of clarity. We hope the new version of the PDF is now available to you.

---

### Official Review · Reviewer_FqNe · 2024-11-08

**Soundness:** 3
**Presentation:** 3
**Contribution:** 2
**Rating:** 6
**Confidence:** 4

**Summary:**

The authors propose an approach for detecting "Shared Decodable Concepts" in human visual fMRI data: clusters in CLIP space that are decodable from common sets of voxels across multiple participants. These clusters are identified using a modified implementation of the DBSCAN clustering algorithm, which identifies clusters in high-dim spaces. In particular, the algorithm operates over vectors of linear decoding weights that map between voxels and CLIP latents. Each weight vector is treated as a brain-decodable concept for a given voxel, such that cosine distance between the weight vectors corresponding to different voxels from different subjects can be used to index the similarity of the concepts encoded therein. After clustering, the authors interpret the SDCs by inspecting a set of "positive" representative images (and words) that are associated with the centroid of each one, and by further identifying groups of "negative" images (and words) that are represented in a maximally different manner. These analyses enable the authors to identify many well-studied aspects of visual cortical organization, such as areas that are tuned to face, body, scene, and word content, and to identify some less-well-studied features such as crowds and lighting. These observations may form the basis for future hypothesis-driven studies that seek to further probe the nuances of the representations of these visual concepts.

**Strengths:**

- The idea of clustering different subjects' parameter/weight vectors as a means of studying what is common (or different) across individuals' brains is a promising direction that merits more attention and exploration in the field. It's a clever way to overcome the challenges of aligning different brains, while respecting the nuances of individuals' representational signatures.
- Beyond the well-studied forms of category selectivity that pop out in the SDC analysis, some of the observations are new and interesting (e.g. a "softness" cluster). Identifying these signatures was only possible because of the sophisticated overall methodological approach applied to a rich dataset. These provide provocative directions for future research, to be sure they are actually replicable signatures across datasets rather than clusters that are biased by the specifics of NSD and/or the use of CLIP.
- The authors make use of the richest possible data (NSD) available, and I appreciate the careful treatment of noise ceiling-related issues in the paper. It's also important that the authors ran a control comparing CLIP decoding to ridge regression.
- It's useful that this work provides further empirical perspective on the hotly debated topic of whether food selectivity is separable from color representation in the ventral stream.
- It's powerful to provide clear visual intuition of both the positive and negative examples that correspond to each cluster, as a way to bolster both the interpretability and the hypothesis space that each SDC provides.
- I deeply appreciate that the authors discussed the possibility of bias in the conclusion, which might arise from the use of NSD stimuli, and which might arise from the use of CLIP.

**Weaknesses:**

- In general the paper does a poor job citing other relevant literature, missing some critical papers on category selectivity in the visual system, as well as on recent efforts to use CLIP models for data-driven interpretability of visual feature tuning. Given the topic of the paper, the fact that it has only 20 citations total is insufficient to cover recent work in this area. For example, the authors should connect their observations of body-related selectivity to other works such as: https://www.nature.com/articles/s41467-020-16846-w, https://www.nature.com/articles/s42256-023-00753-y, and perhaps others.  Citing few works beyond the "original" papers identifying FFA/PPA etc makes it hard to understand how many of the findings are novel versus replicates of other studies. I was also surprised that the authors did not cite/discuss the BrainDIVE paper (https://arxiv.org/pdf/2306.03089) and any other works that use NSD and CLIP to propose new kinds of representational distinctions and signatures in the visual system. Furthermore, Stan Dehaene has relevant work on numerosity representations in the brain that should be cited/discussed. The discussion of food/color representations is missing a key citation from Pennock et al (https://www.cell.com/current-biology/pdf/S0960-9822(22)01893-0.pdf)... and so forth. I'd also recommend discussing existing methods of functional / representational alignment that do not required projecting individuals into a shared anatomical space, such as hyperalignment (Haxby) and shared response modeling (Chen). Why would simpler methods like these be insufficient for this problem of revealing shared dimensions and visualizing the images that project strongly/weakly onto them? I believe that there's extra "juice" to be gained from the CLIP-based approach, it would just help if this historical context were touched on somewhere in the paper.

- There are many ways to achieve high distance from a given cluster centroid in high dimensional space. I understand the reason for looking at the most- and least-associated images with each SDC cluster, but it's not clear that being atypical in a clustering sense implies that those images would actually have a suppressive impact on the voxels' or regions' response profiles. The results here provide clear testable hypotheses, but the impact of the work is limited without further data collection to validate/falsify the positive-negative axes the authors propose.
- The introduction should convey why combining data across participants is a necessary step to discover new forms of category selectivity - this is not obvious. Would a straightforward clustering algorithm applied to the voxel tuning within individual subjects reveal many of these same clusters? The authors do not present analyses that clearly show that cross-subject analysis is necessary to uncover particular forms of interpretable latent dimensions.

- The clarity could have been improved by streamlining the extensive methods description that occurs before any main results are presented, leaving the nitty-gritty details for the supplement. Sections 2 and 3 place a high working memory burden on the reader - reducing this could help a lot with readability.

- This might seem like a minor/pedantic point, but the authors' use of the word "semantic" to describe feature tuning in high-level visual cortex is a bit fraught - even if different regions have selectivities that seam meaningful to the human eye, that does not mean that the neurons themselves necessarily compute semantically interpretable features. (Furthermore, language fMRI researchers likely also have a very different conception of what "semantically tuned patches of cortex" means, compared to vision researchers.) I would recommend that the authors reconsider wordings such as "semantic concepts that drive activity" or "semantically-tuned patches" throughout the paper, in favor of language that describes forms of "visual feature tuning" that may be semantically interpretable, post-hoc, without implying that the underlying cells themselves "know" what semantics are.

**Questions:**

- One main question I am left with is: how many of these proposed clusters contain novel interpretable forms of feature tuning? It is difficult to know without a much more systematic attempt to cite relevant literature on visual category selectivity.
- It remains unclear to me whether cross-subject analysis is required to derive these forms of insight, as discussed above.
- Another key question relates to the issue of bias mentioned above. It would be straightforward to attempt to replicate some of these analyses using BOLD5000, THINGS, or other large datasets to address the possibility of bias that arises from the NSD stimuli. Another critical issue is whether relying on the CLIP feature space unavoidably biases the outcomes of analyses toward semantically interpretable outcomes. In other words, does the use of CLIP cause us to "overfit our understanding" of these brain areas to a feature space that was constrained to be language-aligned? I am not positive how I would go about addressing this issue, but at minimum I would appreciate if the authors attempted to discuss it.
- Given that orientation tuning is a hallmark of V1 representations, why did V1 not show up in the corresponding cluster(s), only V2/V3?

Overall, given the extent of concerns listed above, I am unable to rate the paper higher than a 3 currently. In principle, I am willing to raise my review score if the authors are able to better situate this work in the context of other literature, and more clearly pinpoint the specific novel findings/predictions that differentiate it from other work that has used CLIP to try to understand NSD.

---

> ### Author Response · Authors · 2024-11-20
> **FqNe Response**
>
> We thank the reviewer for kindly taking the time to provide useful feedback on our submission. We plan to split our response across the identified points. We hope this aids the reviewer in considering our response.
>
> Missing Citations
>
> Thank you for the citations, we have incorporated them.  Several of these very relevant citations were genuinely omitted in error as we edited the paper to satisfy the page requirements. These include: Stan Dehaene’s work on numerosity (Dehaene et al., 2003; Dehaene, 2007; Nieder and Dehaene, 2009), the missing Pennock et al. (2023) citation and the BrainDIVE citation. We were honestly also shocked they had been omitted and apologize for the oversight.  Thank you for bringing this to our attention.
>
> Existing Functional Alignment Metrics
>
> SRM / Hyperalignment are functional alignment metrics that require that participants see the same stimuli in the same order, but most NSD samples are images seen by only one participant and very rarely are the same images seen in the same order.  Thus, it’s not possible to apply these to NSD data. We have added additional wording in the submission to better contextualize the choice of clustering algorithm inspiration (DBSCAN) and why other approaches are less suitable. The key aspect of our proposal is that we can map into CLIP (or any other potential model space) and do cross-participant clustering when different stimuli are viewed among the participants. We will update the wording to better reflect this point.
>
> Most- and least-associated images
>
> The reviewer wonders how “being atypical … implies that those images would actually have a suppressive impact on the voxels' or regions' response profiles.”  If one considers the model:
>
> $g^{(k)}(x) = W^{(k)} x = \hat{y}^{(k)}$
>
> $W^{(k)} x = \sum_{i=1}^{v} w^{(k)}_i x_i$
>
> Thus each voxel $x_i$ contributes a multiple of $w^{(k)}_i$ to prediction $\hat{y}$.  If $x_i$ is above baseline, $\hat{y}$ looks more like $w^{(k)}_i$.  If $x_i$ is below baseline, $\hat{y}$ looks more like $-w^{(k)}_i$.
>
> Thus, it’s not that an image is atypical, it’s that the negated images are actually associated with below baseline activation in $x_i$. We’ve used the term suppression in the paper, but will rewrite it to just say reduced activation as we agree that it’s impossible to determine if there is actual suppression at work here.
>
> It is also true that our work can only generate hypotheses, not test them.  We have developed a web-based viewer for our concept clusters that allows for exploration of clusters on flatmaps and the associated images. This hypothesis-generation tool supports the merit of our approach, but could not be released in an anonymous way. We will amend our phrasing to clarify our intentions in this regard. We thank the reviewer for providing this feedback.
> We also agree that running our method on additional datasets could provide more insight.  However, NSD is a remarkable dataset in both scope and data quality.  We chose it amongst the available datasets for that reason and think that part of the power of our results stems from the quality of the data.  It is true, however, that the choice of dataset and CLIP could bias our results. We will add mention of this to the discussion.
>
> Single vs Multi-Subject Analysis
>
> The ability to decode a novel finding in a single participant is interesting, but the question arises: “Is this something unique to this specific participant? Or is it something more fundamental and shared among multiple participants?” Our goal was to avoid participant-idiosyncratic effects. However, a single-subject analysis on the individual NSD subjects is an interesting direction.  Our method could be extended to first cluster on a per-participant basis, and then cluster across participants.  One could then contrast elements that appear in individual subjects to those which appear only when considering multiple subjects.  An interesting direction for future work!
>
> Methods Section
> We have moved Figure 1 to the supplemental information so it reduces the burden on the reader with regard to the methodological details before the results are presented. We hope this renders the reading experience easier for the interested audience and we are glad to receive this helpful feedback.
>
> Updated Terminology
>
> We will update the language in our paper to reflect the point raised by the reviewer and we now reference “semantically interpretable decodable concepts” to avoid potential confusion and misattribution of certain terminological nuances by ICLR’s varied audiences.

---

> > ### Author Response · Authors · 2024-11-20
> > **FqNe Response (additional)**
> >
> > V1 absence of activity (orientations)
> >
> > We thank the reviewer for their observation that V1 was not consistently implicated in the orientation (horizontal / vertical) stimuli. We discussed this point during the write-up of our results. We believe CLIP is sensitive to cohesive, caption-able information, which might be at odds with the orientation column sensitivity of V1 regions. The varying orientations seen in NSD images appear at varying points in the images (e.g. clock towers are not always at the same point in images, images of beaches or airplanes are variably higher or lower and this is detected more robustly as information is pooled together in slightly higher visual regions (V2/V3) but outside of the range of the spatial scale of the orientation columns in very early visual regions. This is why we believe there is a lack of decodable information in V1 using our method.
> >
> > Novelty of Clusters
> >
> > We will update the motivation of our approach particularly with regard to the other work done on NSD as we strongly believe our method has added the methodological toolkit. Our approach is the first one to make use of below-baseline responses to better constrain the potential meanings of the positive responses (e.g. In one cluster where food images being a primary positive cluster concept, we noted the negative images were all black-and-white, leading to our interpretation of the cluster as being color- rather than food-related). In addition, we also introduce a new clustering method that is able to find highly arbitrary cluster shapes (thanks to DBSCAN) reimagined to work with multi-subject data. In addition, we can utilize data when the images are viewed by single participants; we do not require that responses are to shared stimuli (different to other NSD papers). We will update the paper to better reflect these points.

---

> > > ### Comment · Reviewer_FqNe · 2024-12-03
> > >
> > > I appreciate that the authors have put meaningful effort into addressing the feedback I provided during the initial review period.  This has helped clear up several of my earlier confusions. The authors provide helpful clarity regarding existing functional alignment metrics, the fact that least-associated images are interpretable in terms of below-baseline activation, and the absence of orientation clusters in V1.  I am slightly less persuaded by the idea that avoiding subject-idiosyncratic effects should be a "goal" of studies that use NSD, since the dataset is so well suited to study the rich structure that exists within each individual, but this is more a matter of personal philosophy rather than a flaw with the current work. I appreciate that the authors gestured toward more subject-specific extensions of this work as a possible future direction. Overall, the paper narrative feels more streamlined, and a number of key citations have been added.
> > >
> > > Given the above, and because of the authors' good faith effort to address all the reviewers' concerns and make substantive changes to the paper, I am happy to raise my overall score from 3 to 6 and recommend acceptance.

---

### Official Review · Reviewer_oMEW · 2024-11-11

**Soundness:** 2
**Presentation:** 4
**Contribution:** 2
**Rating:** 5
**Confidence:** 4

**Summary:**

This study aims to identify brain areas that respond to specific visual concepts, aiming to expand our understanding of functional localization beyond known areas like the fusiform face area. Using a model based on CLIP embeddings, the authors map brain responses to clusters of shared visual concepts, called "Shared Decodable Concepts" (SDCs), across multiple participants. By applying a modified DBSCAN clustering algorithm, they reveal consistent brain regions linked to both previously known (e.g., faces, places) and novel visual concepts (e.g., specific body parts, numerosity, perspective). This method particularly allows data analysis within each participant’s unique brain structure, avoiding issues with traditional alignment methods.

**Strengths:**

- The discovered novel selectivities observed in the study are quite interesting and potentially good hypotheses for future brain imaging studies
- The perspective that both active and suppressed neural responses are useful for identifying brain regions tied to distinct visual concepts is somewhat unique and not something I've seen in prior studies
- This method circumvents the need to align subjects data, while still being able to extract shared representational structure across participants

**Weaknesses:**

- More methodological details are needed. More motivation behind why DBSCAN clustering used over other clustering algorithms? How do the authors select the epsilon which ultimately determines the number of clusters?
- The authors should acknowledge that there is a lot of human bias involved in interpreting the concepts based on the activated images. For e.g. in 4.1, the authors interpret the negative representative images to indicate images that lack clearly visible faces. However, many other interpretations are also consistent with these sets of images (e.g. they all contain scenes). Similarly Cluster 2 in Fig. 5 could mean something other than bodies too (implied motion?). The subjectivity involved in interpreting the clusters makes me less enthusiastic about the study. Automating this interpretability analyses in future work could be interesting.
- To better understand the strengths of the proposed methodology, it would also be helpful to compare it against natural candidates for understanding shared representational structure across participants, like Canonical Correlation analysis

**Questions:**

The authors frame their study in the context of previous work on identifying functionally localized brain regions. However, it is unclear how their methodology would specifically highlight concepts confined to a few visual areas versus those spread across the brain. Instead, the approach appears to identify concepts that are easily decodable from brain signals across participants, regardless of whether decoding relies on signals from a small localized area or broader, distributed regions.

---

> ### Author Response · Authors · 2024-11-20
> **oMEW Response**
>
> We thank the reviewer for kindly taking the time to provide useful feedback on our submission. we plan to split our response across the identified points. We hope this aids the reviewer in considering our response.
>
> Clustering Motivation
>
> We selected DBSCAN because it can select arbitrary cluster shapes and sizes, bypassing some of the restrictions of other clustering algorithms (e.g. k-means). DBSCAN has been used to model the response profile of V4 neurons in the primate brain. We hypothesized that this flexibility of DBSCAN could improve cross-participant clustering since we do clustering in fMRI-decoded CLIP space (where different dimensions contain potentially very different stimulus properties). Other methods of (functional) alignment require participants to see the same stimuli in the same order and work over common responses to these same stimuli, which precludes these methods from being applied to NSD in which most stimuli are only viewed by individual participants. We will update the paper to better motivate our choice of clustering algorithm and better clarify this point.
>
> Epsilon Value Choice
>
> We examined multiple choices of epsilon values as a way to handle clusters of different density. We suspected that varying this hyperparameter might capture different concept vectors because different brain areas are likely more/less consistent in voxel responses across participants (leading to more/less dense clusters).  The range of values we report were the range for reasonable results.  If epsilon is too low and no clusters are found because there are no core points.  Too high and there are only one or two massive clusters reported.  The range of epsilon that derived interpretable clusters turned out to be fairly tight.
>
> Human Interpretation Subjectivity / Automation
>
> We agree there is a level of human subjectivity involved in interpreting the clusters.  Our approach is a new paradigm for hypothesis generation and this is primarily where we see the benefit of our proposed approach. The potential confounds raised are valid, but until now a system like this (that contrasts two polarities of a concept vector) didn’t exist.  All data-driven analyses are likely to have similar issues with subjectivity.  We agree that future work to automate this analysis is interesting. Perhaps we could use the word clouds to facilitate such an analysis. In addition, this reviewer’s comments actually support the point we make: The positive and negative images provide additional signals as to what concepts might actually represent, leading us towards more clear hypotheses for future testing.
>
> Comparison against other Approaches (i.e. CCA)
>
> CCA is an interesting idea.  However, in NSD there are only a few images shared by all participants so the typical CCA methods (which require the same sample to appear in all sample spaces) wouldn’t apply.  Furthermore, because we are working with weight vectors in CLIP space, all participants are already in the same feature space (one of the advantages of CCA is the ability to combine data in different feature spaces, which isn’t required here).  Thus, we believe the right approach is some kind of clustering.  We chose DBSCAN for the reasons outlined above, but also because it can handle large datasets like NSD.  It also has the notion of core points, which was naturally extendable to take on the participant-neighbor constraint we describe in the paper.  We also experimented with tSNE, but it has its own disadvantages (multiple parameters to tune, results can be unstable).
>
> Our work is quite novel so there aren’t many alternatives to compare to.  Instead, we will add to our discussion some of the other NSD-related papers that have emerged since the submission of our work, and contrast their results to ours. We agree that this addition will strengthen the motivations in the paper.
>
> Distributed vs Localized
>
> The reviewer is correct that the distribution of the voxels across cortex is not a consideration of the clustering algorithm.  If multiple distributed brain areas have the same responses to the stimuli, they will be picked up by our model.  This is because our model only considers the weight vector in CLIP space associated with each voxel.  If those weight vectors are similar (using cosine similarity) the associated voxels will join the same cluster (provided there are voxels from enough participants).  The fact that our method mostly finds areas localized to the visual stream is because of the stimuli used, but also probably also because of the range of epsilons used.
>
> One future direction would be to use a hierarchical DBSCAN which would consider a range of epsilons in the same clustering run, allowing for both more and less dense clusters to be identified.  This may lead to both distributed and localized clusters emerging as the hierarchical clustering proceeds (assuming the distributed clusters are less dense).  We will add this to our future work discussion.

---

### Author Response · Authors · 2024-11-22

We are happy that reviewers agree that the novel findings we observed are interesting and potential avenues for future hypothesis generation (**oMEW**, **FqNe**, **UQYg**).

Reviewers are happy to see the novel applications of our approach, for example in identifying brain regions tied to distinct concepts that are activated and suppressed (**oMEW**,  **LDPq**). Reviewer **FqNe** further points out the relevance of our findings to some hotly debated issues in the field (food concepts and their interaction with color processing).

We thank the reviewers for appreciating our novelty method for cross-subject decoding without aligning brain spaces among subjects (**oMEW**, **FqNe**, **UQYg**), particularly by working in the feature space of CLIP. We agree with reviewer **FqNe** that this exciting opportunity was largely driven due to the accessibility of the large and rich dataset (NSD).

Reviewer **FqNe** praised the attention to detail we strived to get right, particularly in our treatment of comparing our model to ridge regression, noise-ceiling related issues (which reviewer **LDPq** also agreed with) and treatment of potential biases in the analyses arising from the dataset. We also appreciated reviewer **UQYg** in their compliment on our figures.

We thank reviewer **LDPq** for recommending publication of our work at ICLR 2025 and for the kind words supporting the hard work we put into this paper. We are heartened that the care and attention we put into the paper is appreciated by this reviewer.

We also would like to address a common criticism in the lack of relevant citations. We consulted our bibliography and found many of the suggested works listed but uncited. We discovered that during our final re-arrangements of the text body to fit the page requirements, citations were removed from text and not always fully added elsewhere. We were shocked to find this and we apologize that we submitted the paper without the coverage of prior literature we intended. We have taken care to address this point and hope the reviewers now agree that we have corrected this error. Some citations suggested by reviewers genuinely were new to us and we are thankful for them being pointed out and these will be added.

We hope to have satisfactorily answered more reviewer-specific questions in the individual responses below. We would kindly like to ask if the reviewers have any further concerns with which we could possibly help. We would be happy to respond during the remainder of the discussion period.

---

> ### Author Response · Authors · 2024-11-28
> **Interactive Viewer: Screenshots added to Appendix**
>
> To the Appendix we have added screen shots that show our interactive viewer that allows users to explore our results in greater detail.  We have found this viewer to be very helpful as we considered our SDC clusters as it allows one to view both the cortical location of clusters and also the most and least associated images.  We look forward to releasing a public version of the viewer upon paper acceptance.

---

### Meta-Review · Area_Chair_m5qa · 2024-12-19

**Metareview:**

This paper introduces a contrastive model that maps brain responses during naturalistic image viewing to CLIP embeddings, revealing Shared Decodable Concepts (SDCs)—clusters of visual-semantic features linked to common brain regions. Using a novel adaptation of DBSCAN, the study uncovers known and previously unreported visuo-semantic sensitivities, such as distinct food-related clusters and areas tuned for hands/legs or numerosity. This methodology advances the characterization of visuo-semantic representations in the brain by integrating multimodal neural networks with innovative clustering techniques.

All the reviewers agree about the interest and merits of the paper. Substantial improvments have also been approved during the rebuttal period. In this light, I am recommending the paper for acceptance at the ICLR 2025 conference

**Additional Comments On Reviewer Discussion:**

Among other, the paper has been revised in order to incorporate missing related works that were highlighted by reviewers. It turned out that this was an issue in the original submission. Two reviewers changed substantially thier original scores, leading to a clear acceptance path for this paper.

---

### Decision · Program_Chairs · 2025-01-22

Accept (Poster)